# Genetic structure of a germplasm for hybrid breeding in rye (*Secale cereale* L.)

**Nikolaj M. Vendelbo**[1,2]*, **Pernille Sarup**[1], **Jihad Orabi**[1], **Peter S. Kristensen**[3], **Ahmed Jahoor**[1,4]

**1** Nordic Seed A/S, Odder, Denmark, **2** Department of Agroecology, Aarhus University, Slagelse, Denmark, **3** Nordic Seed Germany GmbH, Nienstädt, Germany, **4** Department of Plant Breeding, The Swedish University of Agricultural Sciences, Alnarp, Sweden

\* nive@nordicseed.com

**Data Availability Statement:** All relevant data are within the manuscript and its Supporting Information files.

**Funding:** The research was funded by Innovation Fund Denmark (grant no. 8053-00085B),

## Abstract

Rye (*Secale cereale* L.) responds strongly to changes in heterozygosity with hybrids portraying strong heterosis effect on all developmental and yielding characteristics. In order to achieve the highest potential heterosis effect parental lines must originate from genetically distinct gene pools. Here we report the first comprehensive SNP-based population study of an elite germplasm using fertilization control system for hybrid breeding in rye that is genetically different to the predominating P-type. In total 376 inbred lines from Nordic Seed Germany GmbH were genotyped for 4419 polymorphic SNPs. The aim of this study was to confirm and quantify the genetic separation of parental populations, unveil their genetic characteristics and investigate underlying population structures. Through a palette of complimenting analysis, we confirmed a strong genetic differentiation ($F_{ST} = 0.332$) of parental populations validating the germplasms suitability for hybrid breeding. These were, furthermore, found to diverge considerably in several features with the maternal population portraying a strong population structure characterized by a narrow genetic profile, small effective population size and high genome-wise linkage disequilibrium. We propose that the employed male-sterility system putatively constitutes a population determining parameter by influencing the rate of introducing novel genetic variation to the parental populations. Functional analysis of linkage blocks led to identification of a conserved segment on the distal 4RL chromosomal region annotated to the *Rfp3* male-fertility restoration 'Pampa' type gene. Findings of our study emphasized the immediate value of comprehensive population studies on elite breeding germplasms as a pre-requisite for application of genomic-based breeding techniques, introgression of novel material and to support breeder decision-making.

## Introduction

Rye (*Secale cereale* spp. *cereale*., genome RR, 2n = 2x = 14) is a cereal crop species belonging to the botanical tribe *Triticea* within the grass family *Poaceae* and a close relative to common wheat (*Triticum aestivum* L.) and barley (*Hordeum vulgare* L.) [1]. Rye is believed to have

Pajbjergfoundation, and Nordic Seed A/S. The funder provided support in the form of salaries for authors NMV, PMS, JO, PSK, and, AJ, but did not have any additional role in the study design, data collection and analysis, decision to publish, or preparation of the manuscript. The specific roles of these authors are articulated in the 'author contributions' section.

**Competing interests:** All authors are employees in the plant breeding company Nordic Seed A/S. The employment does not alter the authors' adherence to all of the PLOS ONE policies on sharing data and materials.

evolved as a hitchhiking weed of its progenitor, an annual wild rye (*S. cereale* spp. *vavilovii*) in cultivated fields of founder crops around 12.000 years ago in the Fertile Crescent of the Near East, acquiring key agronomic traits by co-domestication events [2–4]. First evidence of deliberate rye cultivation dates to the European bronze age 5000 years ago and rye has since remained a prominent crop in the temperate northern hemisphere covering 4.12 million hectares in 2018 [5, 6]. Recognized for its tolerance to abiotic and biotic stress, rye exhibits a superior high yield potential under marginal conditions. In mid-19th century, targeted breeding of rye was initiated in Germany, Poland and Russia leading to the generation of secluded germplasms of which the best known are the 'Petkus' and 'Carsten' gene pools [7, 8]. As an allogamous plant species, self-incompatibility has only recently been overcome in cultivated rye after introgression of a dominant self-fertility gene [9]. Breeding lines in rye are therefore predisposed to severe inbreeding depression, when repetitively self-fertilized due to the high load of recessive deleterious mutations accumulated in highly heterozygous species [10]. Inbred rye lines consequently respond very strongly to changes in heterozygosity with hybrids demonstrating a heterosis effect on all developmental and yielding characteristics with *e.g.* grain yield displaying a 110–140% average relative to midparent performance [11–13].With the discovery of cytoplasmic male-sterility (CMS) alleles, breeding effort shifted from open-pollinating varieties (OPVs) to hybrids [14]. Breeding of hybrids in rye rely on a three-way cross of the formula ($A_{CMS}$ x $B_{NRG}$) x R, involving three components, a maternal line carrying the CMS allele, a non-restorer germplasm (NRG) for maintaining the CMS and a restorer (R) pollen father carrying the restorer of fertility (Rf) allele [15]. In order to achieve the highest potential of heterosis effect, parental lines should originate from genetically separate, heterotic gene pools [16]. In recent years, technological and scientific advances have expedited the implementation of genome-based breeding strategies in rye for a more efficient exploration of the genetic potential [17, 18]. Implementation of marker assisted selection and genomic selection relying on molecular DNA markers has enhanced the selection efficiency leading to increased genetic gains surpassing traditional breeding methods [12, 13, 19]. In 2011 Haseneyer *et al*. [20] published a 5K single nucleotide polymorphism (SNP) array for rye, succeeded by the 600K array by Bauer *et al*. [1]. Recently Rabanus-Wallace *et al*. [21] published the first chromosome-scale assembly of the 7.9 Gbp rye genome. In conjunction, these genomic resources represent significant milestones, providing indispensable tools for elucidating genes underlying important agronomic traits, implementation of genomic-based breeding techniques and dissecting population genetics in rye [22]. Through a comprehensive understanding of the genetic architecture, wild relatives, OPVs, and landraces can be efficiently exploited through selected introgression, broadening the genetic basis of elite breeding germplasm and providing access to a rich reservoir of genetic diversity for ensuring continued genetic gains in hybrid breeding [16, 23, 24]. SNP arrays constitute a high-throughput genotyping platform readily implemented by breeding programs for assaying the genetic variation and identifying trait-linked markers by genome-wide association studies for marker assisted selection in elite breeding germplasm. Numerous population studies have been conducted on the *Secale* genus, open-pollinating rye varieties and inbred populations using simple sequence repeats (SSR), diversity array technology (DArT), and, random amplification of polymorphic DNA (RAPD) marker systems [25–27]. In the extensive study by Bauer *et al*. [1], they validated their 600K SNP array in a smaller population study on a diverse panel of wild *Secale* sp. accessions, including inbred lines from heterotic gene pools.

In this study, we report a comprehensive SNP-based study of genetic diversity and population structure in a hybrid rye elite breeding germplasm as prerequisite for application of genomic-based breeding techniques, introgression of novel material and to support breeder decision-making. Our aim was to I) confirm and quantify the genetic separation between

parental populations, II) gain a comprehensive understanding of the genetic characteristics, features and architecture of the parental populations, and, III) Investigate underlying sub-population structures.

## Materials & methods

### Plant material

In total 376 Nordic Seed A/S inbred hybrid rye (*Secale cereale* L.) elite breeding component lines were selected for the study, comprising 250 restorer, 119 non-restorer germplasm (NRG) and 7 cytoplasmic male-sterile (CMS) lines. The restorer lines originated from the 'Petkus' and 'Carsten' gene pools with a predominance of the latter suggested by available pedigree data, whereas information on the precise origin of the NRG lines is not-existent. The CMS male sterility is based on the 'Gülzow' (G) type cytoplasm originating from the German population rye variety 'Schlägler alt' [28].

### DNA extraction

Four seeds of each line were sown in 104 hole plates (51.5 x 31 x 4.5cm), containing fine-grain sphagnum substrate. Plants were cultivated at Nordic Seed A/S greenhouse facilities under natural light and manual irrigation, with night temperatures of 14–16°C and day temperatures of 18–24°C. Seven days after sowing, the lowest section of two coleoptiles and primary leaves were cut, equivalent to 75 mg plant material, and placed in a 96-well Micro-Dilution Tube System (STARLAB International GmbH) containing two 4mm glass beads per 1.2 mL tubes. Plant tissue samples were stored at -20°C for two days prior to freeze drying for an additional two days in a 9L Coolsafe[TM] (LaboGene) apparatus. Preceding the DNA extraction, leaves were crushed at 4000 RPM using a TissueLyser II (Qiagen[®]) bead mill, and DNA extracted using an SDS-based extraction method. DNA concentration and 260/280nm ratio of samples were measured using Epoch[TM] microplate spectrophotometer (Biotek[®]) and quality, *i.e.* evidence of fragmentation by size-visualization on a 1.2% agarose gel.

### Molecular marker resources and SNP genotyping

Samples containing 100 ng and 1.7–1.9 260/280nm ratio long-stranded DNA were sent for single nucleotide polymorphism (SNP) genotyping at TraitGenetics GmbH using a pre-designed Illumina Infinium 15K$_{wheat}$ and 5K$_{Rye}$ SNP iSelect ultra HD chip. Custom rye specific SNPs were comprised of 2698 markers from the 5K array by Haseneyer *et al.* [20] denoted as '5K-set' and 2059 markers from the 600K array by Bauer *et al.* [1] denoted as '600K-set'. Wheat SNPs comprised of 12908 markers originating from the 90K array by Wang *et al.* [29] denoted 'wheat-set'. The markers were prior to analysis filtered for marker allele frequency $\geq 0.005$, missing individual score $\leq 0.2$, and, missing marker score of $\leq 0.1$.

### Data analysis

Population genetic analysis of SNP marker data was done in R studio (v. 1.1.463) interface in R statistical software (v. 3.6.3) by application of various predesigned packages [30, 31]. Marker minor allele frequency (MAF), polymorphic information content (PIC), and, effective population size ($N_e$) was calculated using SnpReady (v. 0.9.6) R package [32]. Calculation of the genetic characteristics of accessions, *i.e.* observed heterozygosity ($H_o$), within population gene diversity ($H_s$), and Wrights F statistics, inbreeding coefficient ($F_{IS}$) and fixation indices ($F_{ST}$) per population and identified sub-populations was done using Hierfstat (v. 0.04–22) R package [33, 34]. Parental pool differentiating SNPs were identified by calculating the marker $F_{ST}$ using

Pegas (v. 0.13) R package [35]. Principal component analysis (PCA) was conducted using inherent functions in R with 3D PCA constructed using rgl (v. 0.100.50) R package [36]. Shared ancestry between accessions were estimated by Admixture model using STRUCTURE (v. 2.3.4) software, assigning each locus probabilistically to an ancestral founder (or subpopulation) K per breeding line on basis of allele frequencies [37]. The model was run with burning period set to 200.000 and Markov chain Monte Carlo to 400.000 iterations, with the results visualized in R studio. Estimate of the most likely number of K was done using the DeltaK method by Evanno *et al.* [38] in Structure Harvester (v. 0.6.94) at K set to 1 to 5 with 5 replicates per level [39]. The analysis was conducted on the complete germplasm to investigate the number of genetically distinct ancestral founders and for each of the parental populations to identify the most likely number of subpopulations. Phylogenetic analysis comprised of a neighbor-joining clustering of breeding lines with Euclidean genetic distance measure using ape (v. 5.3) R package [40]. The tree was constructed after 10.000 bootstrapping iterations with weak nodes (80% recurrence) collapsed into multifurcations and generated using iTOL (v. 5) online tool (http://itol.embl.de/) [41]. Fisher's exact test was conducted to test whether the structural composition, *i.e.* no. of 'clades' and 'singletons' in the phylogenetic tree diverged between populations using R inherent functions. Analysis of pairwise linkage disequilibrium (LD) in the parental populations was done using SnpStats (v. 1.36.0) R package with depth set to 6 and LD estimated as the coefficient of determination ($r^2$) [42]. Prior to analysis, a filtration step was included to remove non-informative population-specific monomorphic markers. Heatmap of the pairwise LD was constructed using LDheatmap (v. 0.99–7) R package [43]. Genetic map of markers and visualization of linkage decay was done using Sommer (v. 2.9) R package with filtration for marker correlation ($r^2$) significance at p<0.0001 [44]. The intersection of the LOESS curve fit to the baseline of population-specific critical value of $r^2$, *i.e.* the estimated linkage disequilibrium (LD) background was set as the delimiter of the linkage decay interval [45]. LD and rate of decay was calculated per parental population the 5K-set and 600K-set markers separately due to divergence in the arrays genetic distance measure.

## Results

### Molecular marker resources and SNP genotyping

Quality filtration of markers for low minor allele frequency (MAF), missing marker, and missing individual across the entire panel led to the identification of 4419 high-quality SNP markers distributed evenly amongst the seven rye chromosomes. The filtered marker panel comprised of 2364 5K-set markers, 1629 600K-set markers, and 426 wheat-set markers. In terms of marker-set performance and informativeness across the entire germplasm a mean MAF of 0.254, 0.250, 0.201 and mean PIC of 0.268, 0.262, and 0.225 was determined the 5K-set, 600K-set and wheat-set markers respectively. Mean MAF was 0.188 across the markers in the NRG&CMS population, omitting 1870 monomorphic markers, with ~50% of the polymorphic markers exhibiting a MAF ≤ 0.1 (Fig 1A). The restorer population portrayed a homogeneous distribution of MAF frequencies with a mean of 0.251, omitting only 38 monomorphic markers. Quality filtration was likewise conducted on the separate parental populations to determine the effect of population size difference. The test yielded an additional 82 high-quality markers for the NRG&CMS population, while 47 for the restorer population. Mean polymorphic information content (PIC) was 0.202 in the NRG&CMS post filtration of monomorphic markers with ≈ 35% exhibiting a PIC ≤ 0.1 (Fig 1B). In the restorer population, mean PIC across polymorphic markers was 0.261, with ≈ 50% exhibiting a PIC value between 0.3 and 0.4. Overall mean PIC was estimated to 0.261.

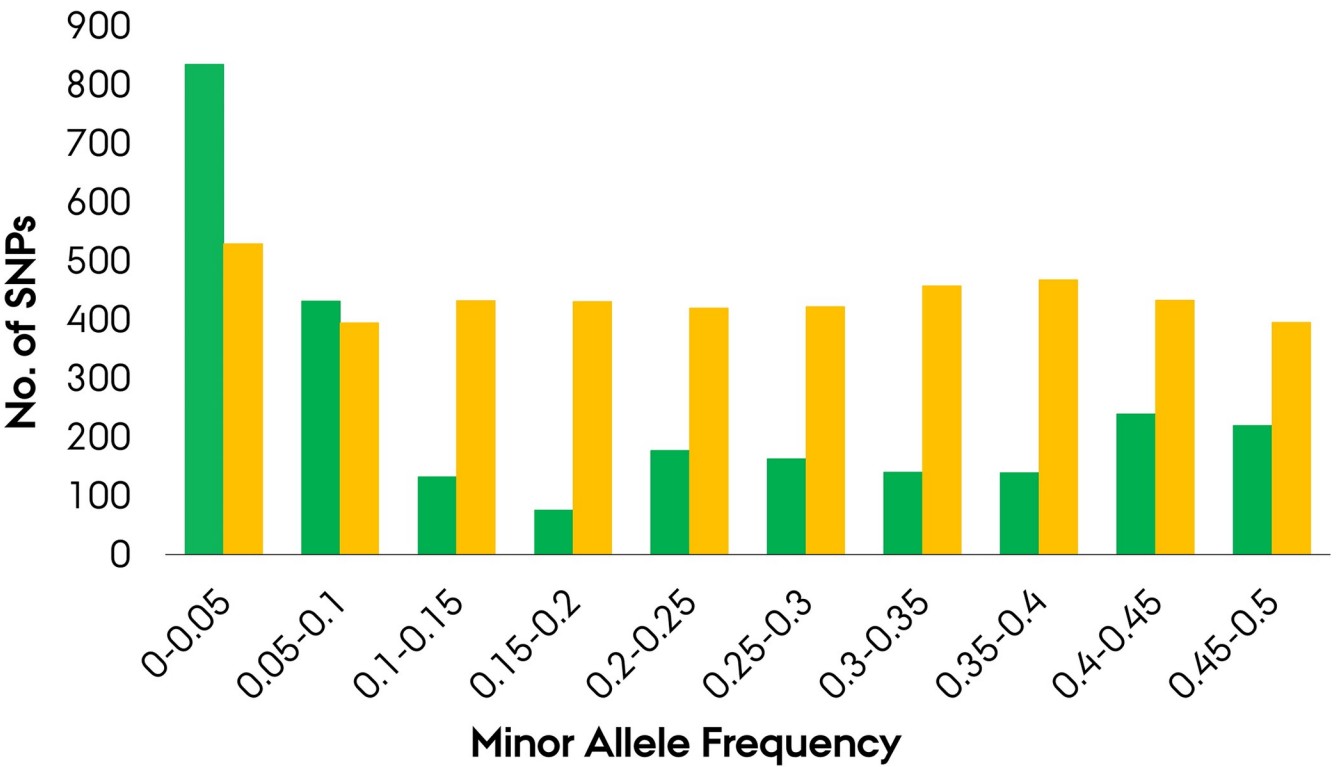

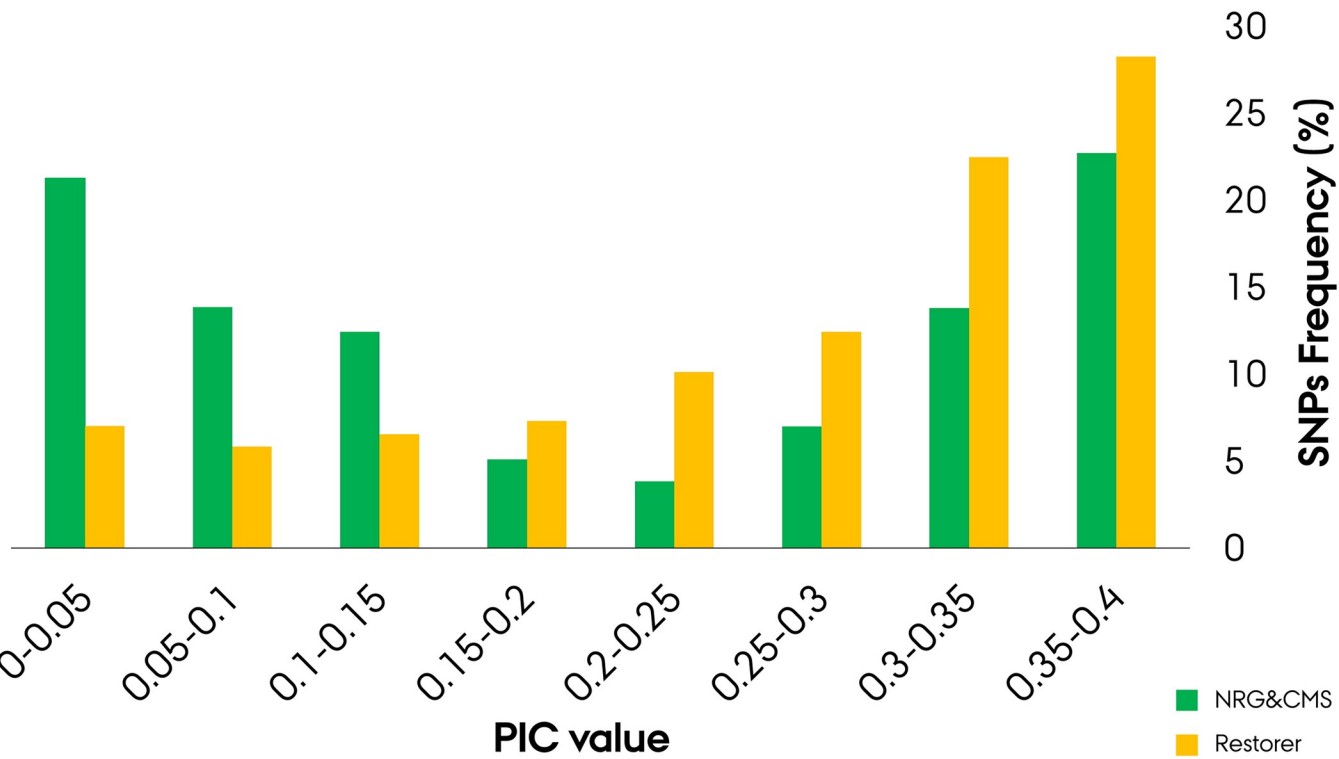

**Fig 1.** Population-wise distribution of minor allele frequency (MAF) (**A**) and polymorphic information content (PIC) (**B**) for 4419 SNP markers in the Nordic Seed elite hybrid rye breeding germplasm comprising a seed mother (NRG&CMS) and pollen father (Restorer) population.

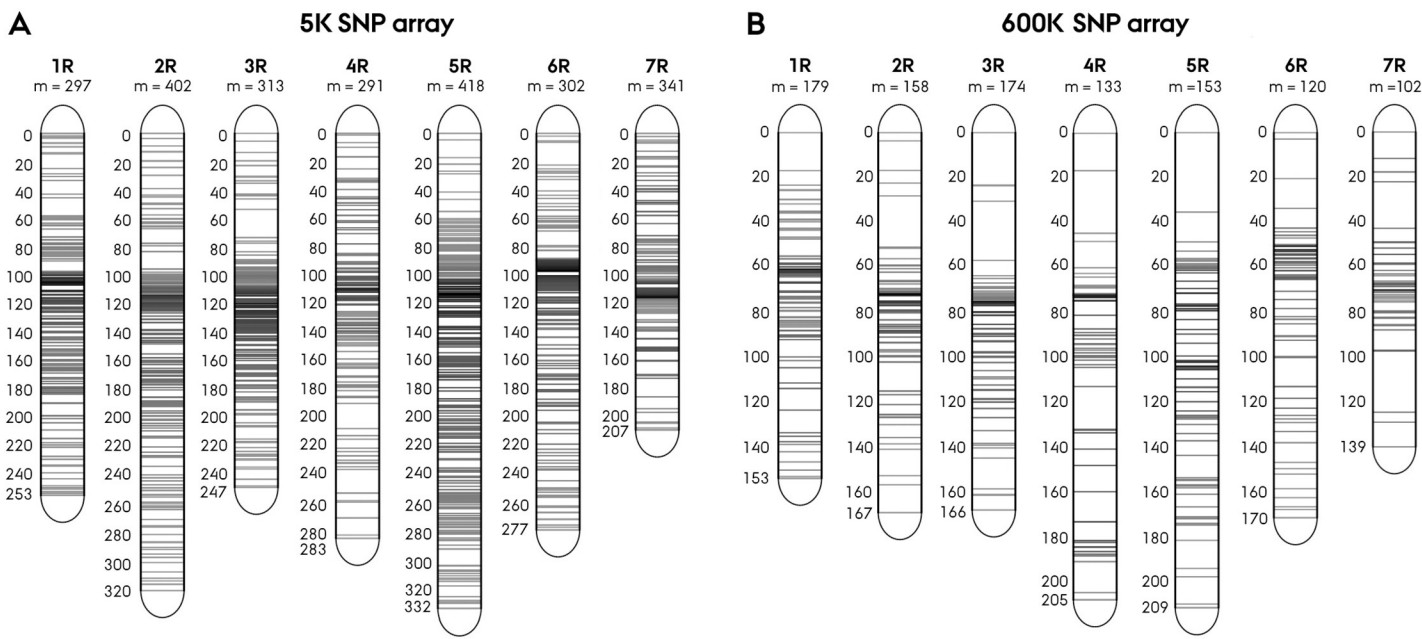

**Fig 2.** Genetic map of SNP markers in rye. **A)** 2364 markers from the 5K array and **B)** 1019 from the 600K array.

Genetic mapping of the 3383 mapped markers revealed an excess of markers mapping to the peri-centromeric region in both the 5K-set and 600K-set markers, increasing the inter-loci distance at the proximal and distal region (Fig 2A and 2B). None of the wheat-set markers had a mapping position in the rye genome.

## Genetic characteristics

Calculation of fundamental genetic characteristics led to the initial discovery of 11 lines portraying evidence of an inadvertent cross fertilization event between the two parental populations. All these 11 lines displayed a high proportion of the opposing ancestral component in an analysis of inferred ancestry at K set to 2 with a mean of 56%. Moreover, 4 lines displayed a mean residual observed heterozygosity ($H_o$) of 41%, suggesting a recent inadvertent cross fertilization (S1 Table). Together, these results led to the decision to discard these lines from further analysis leaving a trimmed population comprised of 242 restorer, 116 NRG and 7 CMS component lines.

In the remaining trimmed population, the NRG&CMS population displayed a mean $H_o$ of 3.6%, within population gene diversity ($H_s$) of 0.250, inbreeding coefficient ($F_{IS}$) of 0.852, and effective population size ($N_e$) of 72, equivalent to a $N_e/N$ of 0.587. The restorer population displayed a $H_o$ of 3.8%, $H_s$ of 0.333, $F_{IS}$ of 0.886, and, $N_e$ of 137 equivalent to a $N_e/N$ of 0.565. Fixation indices ($F_{ST}$) between the two populations was estimated to 0.332.

## Principal component analysis

Initial analysis conducted on the complete set of 376 Nordic Seed hybrid rye elite breeding component lines comprised of a principal component (PC) analysis plot visualizing the genetic architecture across various PC combinations (Fig 3). While the analysis provides an estimate of the proportion of variation explained by the individual PCs, the PCA primarily served in our study to get a firsthand visual impression of the genetic diversity and architecture within

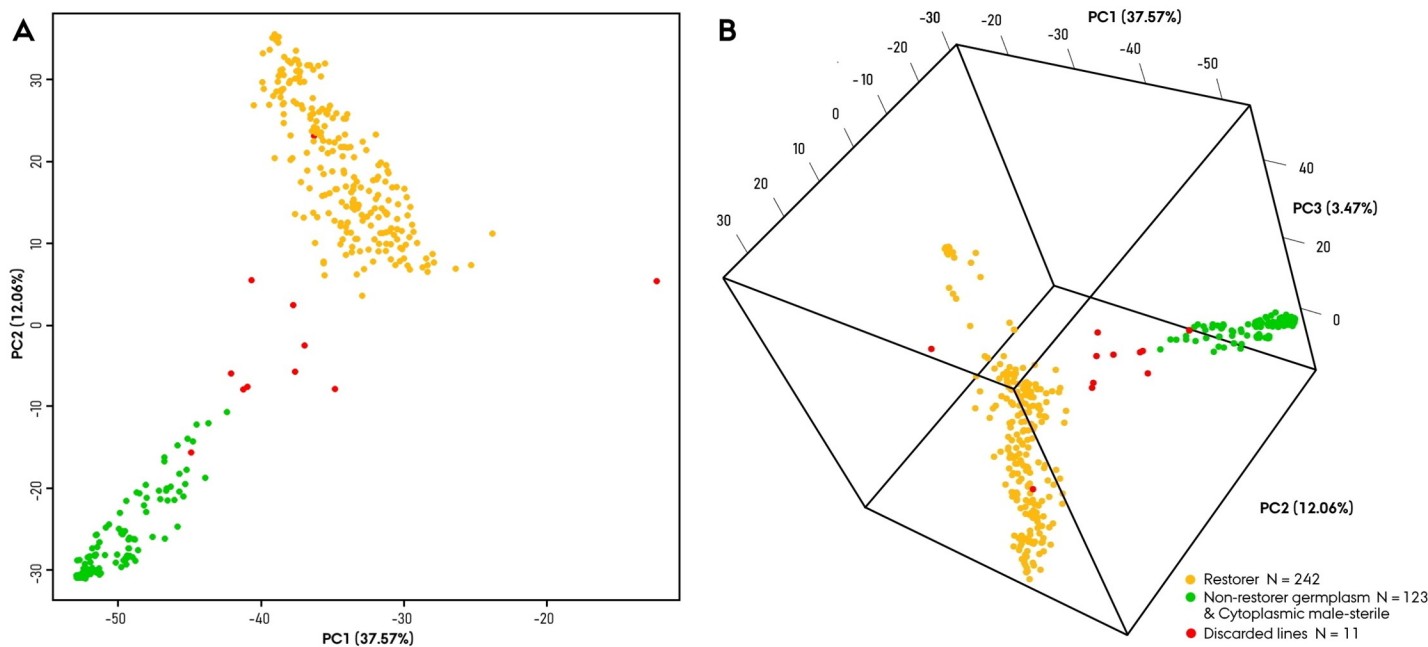

**Fig 3.** Principal component analysis of the Nordic Seed hybrid rye elite breeding germplasm ($N_{pop}$ = 376), based on 4419 SNP markers, (**A**) 2-D plot of principal component (PC) 1 and 2, (**B**) 3-D plot of PC1, 2 and 3. Red dots indicate discarded lines.

the material. In the constructed 2-D PCA plot depicting PC1 (37.57%) and PC2 (12.06%), lines were observed to group into two distinct clusters in accordance to their population, restorer and NRG&CMS (Fig 3A). With addition of PC3 (3.47%) a 3-dimensional PCA plot led to further unfolding of the restorer populations genetic architecture, while the NRG&CMS population remained closely grouped. Whilst informative as a static graph, visualization of the PC in a 3D interactive software facilitates a more comprehensive mining of data structures. Lastly, the PCA, furthermore, served to validate the discontinuation of 11 lines with majority of these observed in the intermediary space diluting the distinct genetic separation between two parental population.

## Phylogenetic analysis

Hierarchical clustering analysis of the trimmed Nordic Seed hybrid rye elite breeding population supported the distinct genetic separation of the two parental populations visualized in a circularized neighbor-joining dendrogram (Fig 4). In order to validate the phylogenetic tree, weak nodes showing less than 80% recurrence were collapsed into multifurcations after 10.000 bootstrapping iterations. This stringent validation-step dramatically reduced the number of inconsistent family clades within the dendrogram, and hence the overall structural complexity in the germplasm. In the constructed neighbor-joining dendrogram the populations were found to exhibit a significantly different structural composition (p = 0.028) in a Fisher's exact test. The NRG&CMS population portrayed 25 clades comprising on average of 5 lines and 11 singletons relative to the restorer population that portrayed 41 clades of similar mean size and 51 singletons.

## Inferred ancestry & sub-population discovery

In order to dissect the underlying sub-population structure within the trimmed elite Nordic Seed hybrid rye breeding germplasm, an Admixture analysis of inferred ancestry was

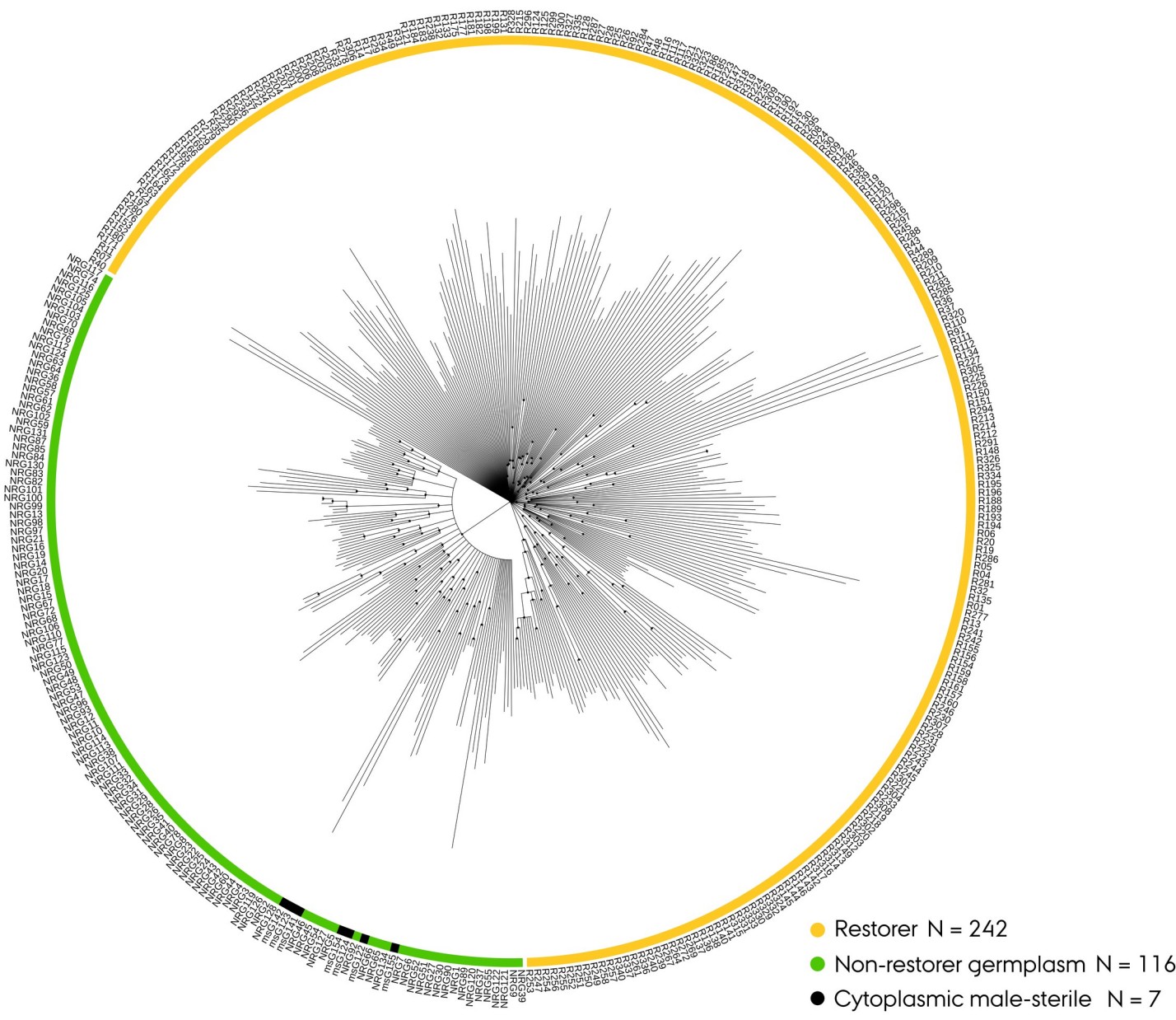

**Fig 4. Circularized neighbor-joining dendrogram of the trimmed Nordic Seed hybrid rye elite breeding germplasm (N$_{pop}$ = 365) based on 4419 SNP markers, after bootstrapping with recurring nodes shown as black dots.**

conducted. This model-based clustering method assigned individual lines based on their allele frequencies to K clusters, *i.e.* ancestral founder populations or sub-populations. Output of the admixture model is a frequency distribution of K ancestral sub-population components for each component breeding line easily presented in a bar plot and colorized according to K clusters (Fig 5). Component breeding lines were aligned in accordance to their phylogenetic relationship in order to support the identification of underlying sub-population structures (Fig 5A and 5F). First step was to validate the existence of two genetically distinct ancestral founders which the parental populations originate by conducting the Admixture model on the entire trimmed population at K set to 2 (Fig 5B and 5G). This was done using Evanno's DeltaK method confirming a historical separation of the two parental gene pools (ΔK = 2) with the

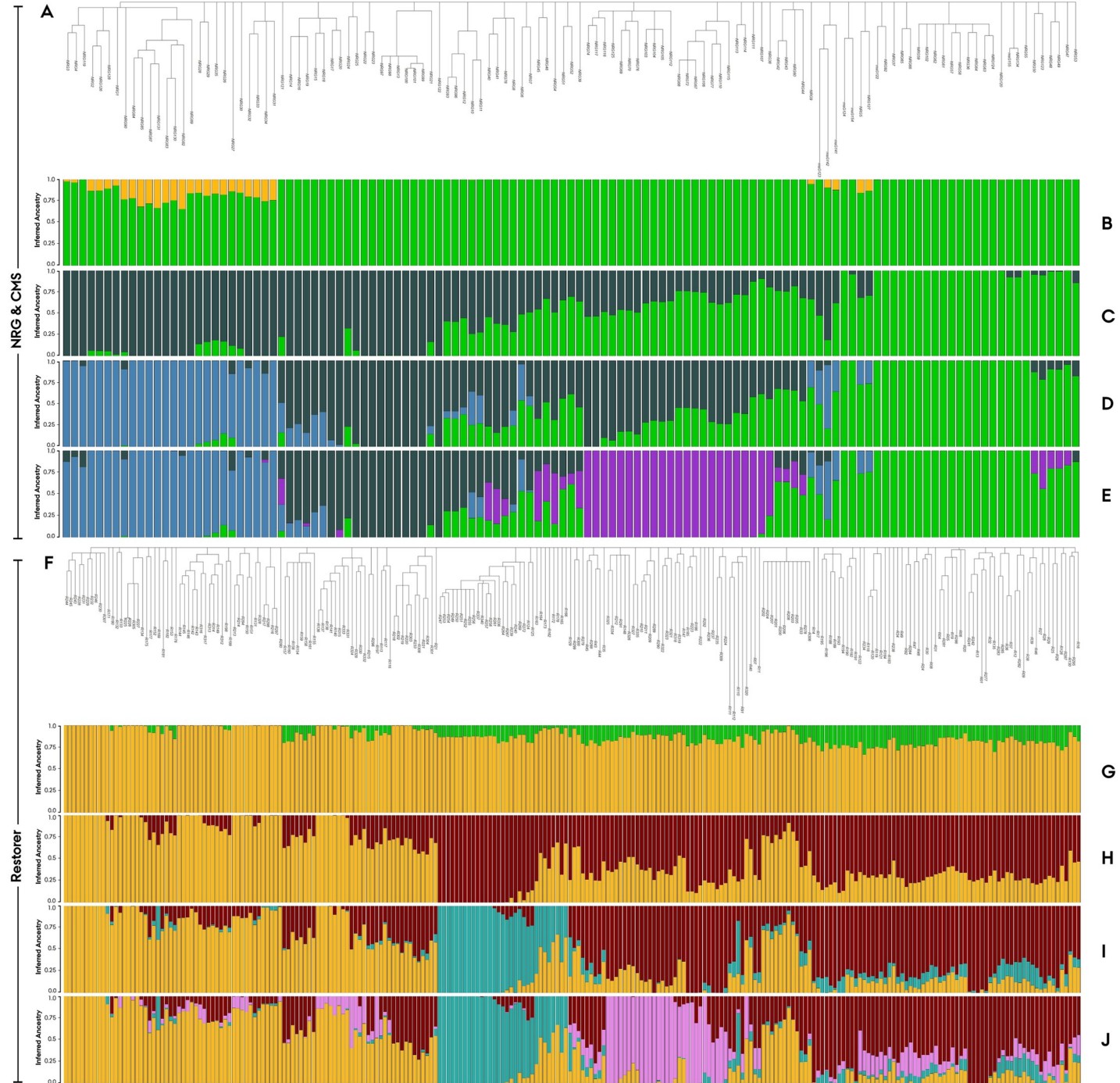

**Fig 5. Admixture model of inferred ancestry and neighbor-joining dendrogram of the trimmed Nordic Seed hybrid rye elite breeding germplasm based on 4419 SNP markers.** Neighbor-joining dendrogram of the parental populations, (**A**) Seed parent (NRG&CMS, N = 123), and, (**F**) Pollen parent (R, N = 242). Analysis of co-ancestry between parental populations (**B**) NRG&CMS (green) and (**G**) R population (yellow) at K set to 2. Discovery of sub-population structure at K set to 2, 3, 4 respectively for the (**C,D,E**) NRG&CMS, and (**H,I,J**) Restorer lines. Colors define fraction of inferred ancestral sub-population.

Admixture model revealing limited extend of co-ancestry. The NRG&CMS population exhibited a mean co-ancestry of 4.3% shared with the restorer population across a subset of admixed lines (Fig 5B), whereas the restorer population exhibited a mean co-ancestry of 11.6% spread

**Table 1. Pairwise fixation indices ($F_{ST}$), observed heterozygosity ($H_o$), within subpopulation gene diversity ($H_s$), and inbreeding coefficient ($F_{IS}$) of parental subpopulations (SP) within the Nordic Seed hybrid rye elite breeding germplasm (A) seed mother (NRG&CMS, N = 123), and (B) pollen father (Restorer, N = 242).** SP constitutes n lines with ≥ 0.95 Admixture inferred ancestry at K inferred subpopulations.

| | A | Non-restorer germplasm & Cytoplasmic male-sterile | | | | | | | | B | Restorer | | | | | | | |
|---|---|---|---|---|---|---|---|---|---|---|---|---|---|---|---|---|---|---|
| | | Pairwise $F_{ST}$ | | | | Genetic characteristics | | | | | Pairwise $F_{ST}$ | | | | Genetic characteristics | | | |
| Inferred subpopulations (K) | K = 4 | SP1 | SP2 | SP3 | SP4 | n | $H_o$ | $H_s$ | $F_{IS}$ | | SP1 | SP2 | SP3 | SP4 | n | $H_o$ | $H_s$ | $F_{IS}$ |
| | SP1 | 0 | | | | 21 | 0.016 | 0.054 | 0.706 | SP1 | 0 | | | | 15 | 0.021 | 0.106 | 0.801 |
| | SP2 | 0.347 | 0 | | | 16 | 0.025 | 0.133 | 0.812 | SP2 | 0.694 | 0 | | | 14 | 0.012 | 0.033 | 0.645 |
| | SP3 | 0.435 | 0.490 | 0 | | 22 | 0.016 | 0.034 | 0.524 | SP3 | 0.437 | 0.594 | 0 | | 11 | 0.034 | 0.198 | 0.824 |
| | SP4 | 0.485 | 0.555 | 0.489 | 0 | 11 | 0.005 | 0.046 | 0.900 | SP4 | 0.420 | 0.570 | 0.304 | 0 | 7 | 0.084 | 0.221 | 0.622 |
| | K = 32 | SP1 | SP2 | SP3 | | n | $H_o$ | $H_s$ | $F_{IS}$ | | SP1 | SP2 | SP3 | | n | $H_o$ | $H_s$ | $F_{IS}$ |
| | SP1 | 0 | | | | 22 | 0.016 | 0.055 | 0.716 | SP1 | 0 | | | | 26 | 0.018 | 0.164 | 0.893 |
| | SP2 | 0.492 | 0 | | | 18 | 0.024 | 0.126 | 0.806 | SP2 | 0.524 | 0 | | | 14 | 0.012 | 0.033 | 0.645 |
| | SP3 | 0.514 | 0.323 | 0 | | 13 | 0.004 | 0.058 | 0.931 | SP3 | 0.293 | 0.484 | 0 | | 13 | 0.058 | 0.261 | 0.780 |
| | K = 2 | SP1 | SP2 | | | n | $H_o$ | $H_s$ | $F_{IS}$ | | SP1 | SP2 | | | n | $H_o$ | $H_s$ | $F_{IS}$ |
| | SP1 | 0 | 0.560 | | | 23 | 0.017 | 0.055 | 0.691 | SP1 | 0 | 0.436 | | | 41 | 0.017 | 0.185 | 0.906 |
| | SP2 | 0.560 | 0 | | | 33 | 0.020 | 0.182 | 0.842 | SP2 | 0.436 | 0 | | | 19 | 0.014 | 0.062 | 0.778 |

more homogeneously across the entire panel (Fig 5G). Evanno's DeltaK method was furthermore applied to determine the most likely estimated number of founders for the individual parental populations. Analysis suggested 3 ancestral founders in both the NRG&CMS and restorer population. To examine the underlying sub-population structure, K was set to 2, 3, and 4 within the individual populations (Fig 5C–5E and 5H–5J). For each of the identified subpopulations, $H_o$, $H_e$, $F_{IS}$ and pairwise $F_{ST}$ was estimated to dissect their genetic characteristics and differentiation (Table 1). Breeding lines portraying a predominance of a single ancestral subpopulation component (≥ 0.95) was found to cluster according to their position within the neighbor-joining dendrogram. Despite considerable divergence in parental population size the number of lines assigned to subpopulations were predominantly similar, indicative of a more admixed, genetically complex population structure within the restorer.

## Linkage disequilibrium & decay

Non-random association of alleles at different loci, *i.e.* linkage disequilibrium (LD) and the decay of linkage are important genetic characteristics of breeding germplasms providing insight into the marker density essential for trait-linked marker discovery by association genetic studies. In our study, the intra- and inter-chromosomal, *i.e.* genome-wide, LD was calculated for the two populations using 3383 mapped markers comprised of 2364 5K-set markers and 1019 600K-set markers (Table 2). Filtration for population-specific monomorphic markers led to the total reduction of 1400 and 85 mapped markers in the NRG&CMS and restorer population, respectively. While the linkage analysis had to be conducted for the individual 5K-set and 600K-set markers due to divergence in the arrays genetic distance measure, the two marker sets led to highly similar estimates of LD (Table 2). The NRG&CMS population exhibited a mean genome-wise LD of 0.394 and 0.364 for the 5K-set and 600K-set markers respectively with a mean of 0.385 across the entire marker panel. Restorer population exhibited a mean genome-wise LD of 0.150 and 0.204 for the 5K-set and 600K-set markers respectively with a mean of 0.166 across the entire marker panel.

Comparison of the intrachromosomal LD between populations led to the finding of a similar lower mean LD on the 4R chromosome, while higher level on the 3R chromosome (Fig 6, Table 2). The restorer population distinguished itself by a ≥15% relative lower and higher LD

**Table 2. Mean pairwise intrachromosomal linkage disequilibrium in the Nordic Seed hybrid rye elite breeding component germplasm, comprising a seed mother (NRG&CMS, N = 123) and pollen father (R, N = 242) population using 3383 mapped SNP markers originating from the 5K array and 600K array.** Prior to analysis quality control (QC) was done to remove population-specific monomorphic markers.

| Chromosome | | | | 1R | 2R | 3R | 4R | 5R | 6R | 7R | Mean | Total |
|---|---|---|---|---|---|---|---|---|---|---|---|---|
| **Chromosome length (cM)** | | | 5K | 253 | 320 | 247 | 283 | 332 | 277 | 207 | 274 | 1919 |
| | | | 600K | 153 | 167 | 166 | 205 | 209 | 170 | 139 | 172 | 1209 |
| **No. SNP Markers** | Total | | 5K | 297 | 402 | 313 | 291 | 418 | 302 | 341 | 378 | 2364 |
| | | | 600K | 179 | 158 | 174 | 133 | 153 | 120 | 102 | 146 | 1019 |
| **QC SNP Markers** | NRG&CMS | | All | 235 | 415 | 156 | 274 | 350 | 281 | 271 | 283 | 1982 |
| | | | 5K | 151 | 293 | 108 | 196 | 264 | 199 | 209 | 203 | 1420 |
| | | | 600K | 84 | 122 | 48 | 78 | 86 | 82 | 62 | 80 | 562 |
| | R | | All | 476 | 559 | 485 | 423 | 561 | 418 | 441 | 480 | 3363 |
| | | | 5K | 297 | 401 | 312 | 291 | 415 | 298 | 339 | 336 | 2353 |
| | | | 600K | 179 | 158 | 173 | 132 | 146 | 120 | 102 | 144 | 1010 |
| **Linkage disequilibrium + SD** | NRG&CMS | | All | 0.322+0.24 | 0.367+0.25 | 0.424+0.29 | 0.319+0.20 | 0.389+0.26 | 0.300+0.19 | 0.420+0.25 | 0.385+0.26 | - |
| | | | 5K | 0.396+0.30 | 0.375+0.24 | 0.546+0.36 | 0.328+0.21 | 0.422+0.27 | 0.326+0.19 | 0.456+0.25 | 0.394+0.26 | - |
| | | | 600K | 0.322+0.23 | 0.388+0.26 | 0.424+0.37 | 0.385+0.23 | 0.441+0.28 | 0.295+0.19 | 0.391+0.24 | 0.364+0.26 | - |
| | R | | All | 0.151+0.17 | 0.134+0.13 | 0.191+0.19 | 0.131+0.13 | 0.153+0.15 | 0.190+0.20 | 0.162+0.14 | 0.166+0.16 | - |
| | | | 5K | 0.146+0.15 | 0.122+0.11 | 0.152+0.13 | 0.129+0.12 | 0.163+0.15 | 0.181+0.18 | 0.167+0.14 | 0.150+0.14 | - |
| | | | 600K | 0.194+0.20 | 0.175+0.17 | 0.295+0.25 | 0.151+0.14 | 0.184+0.19 | 0.267+0.27 | 0.169+0.13 | 0.204+0.20 | - |

on the 2R and 6R chromosome, respectively. The NRG&CMS population was found to exhibit a ≥15% relative lower LD on the 1R and 6R chromosome, while a ≥15% relative higher on the 7R chromosome.

To identify structural differences and patterns between populations, the pairwise intra-chromosomal LD was visualized by a heatmap (S1 Fig). Due to the non-homogeneous distribution of markers, extrapolation of intra-chromosomal distance is not feasible on the heatmap, instead depicting the immediate pairwise relationships. Heatmaps constructed on basis of separate arrays were found to produce comparable intra-chromosomal patterns revealing large LD blocks in the NRG&CMS population with an excess of LD in the peri-centromeric region (Fig 7A, 7C and 7E). The restorer population portrayed a more homogeneously distributed LD

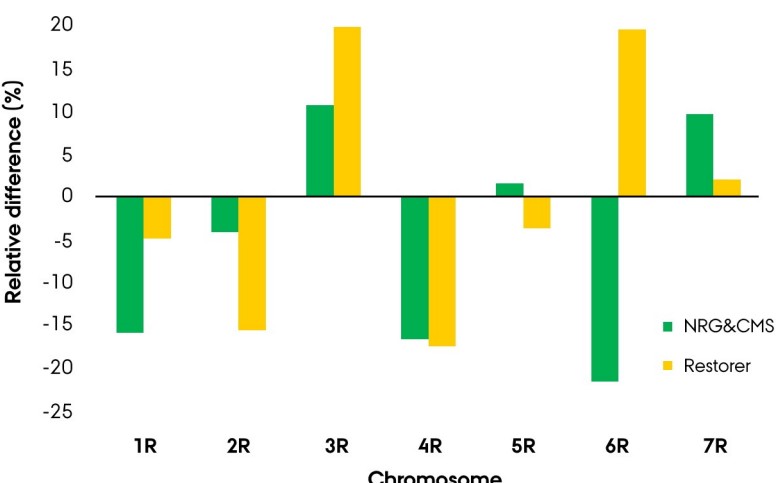

**Fig 6. Relative difference in intrachromosomal linkage disequilibrium (%) compared to the genome-wise mean for the seed mother (NRG&CMS) or pollen father (R) population in the Nordic Seed elite hybrid rye breeding germplasm using 3383 SNP markers.**

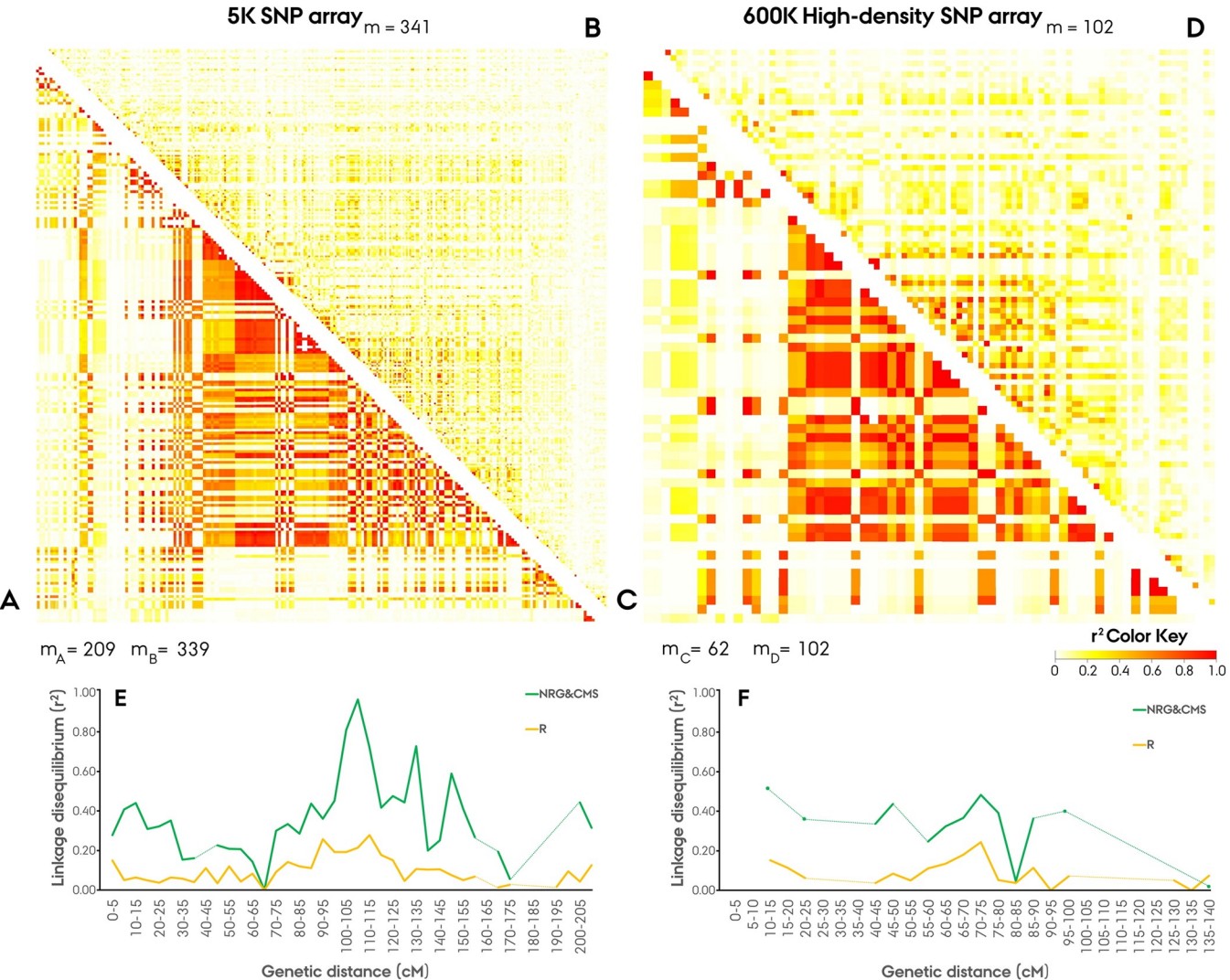

**Fig 7.** Pairwise linkage disequilibrium (LD, $r^2$) across the 7R rye chromosome in the Nordic Seed hybrid rye elite breeding populations using population-specific polymorphic SNP markers (m), originating from a 5K (**A,B,E**) and 600K array (**C,D,F**), visualized by heatmap for the, **A/C**) Seed mother (NRG&CMS), **B/D**) Pollen father (R) population, with the genome-wise distribution of LD across arrays visualized in bins of 5 cM. Missing bins is represented by dotted lines.

with a less distinct pattern of excess LD in the peri-centromeric region and fewer minor LD blocks (Fig 7B, 7D, 7E and 7F). In both populations, the 3R chromosome demonstrated the highest mean LD with this excess readily event in the NRG&CMS population portraying a large LD block spanning from 76 to 108 cM in the 5K array and 76 to 99 cM in the 600K array (Table 2, S1 Fig). In the distal region of the 4RL chromosome, a distinct LD block was present in both populations (S1 Fig). In the NRG&CMS population, the block spanned 12 markers from 184 to 189 cM on the 600K array exhibited a mean pairwise LD of 0.91 and mean marker $F_{ST}$ of 0.54. In the restorer population, the block comprised 5 markers at 189 cM on the 600K array, exhibiting a mean pairwise LD of 0.80 and mean $F_{ST}$ of 0.62. The two populations diverged singularly by more than 40% in relative population-wise mean LD on the 6R chromosome with the increased LD in the restorer population organized in multiple moderate blocks (Fig 6, S1 Fig).

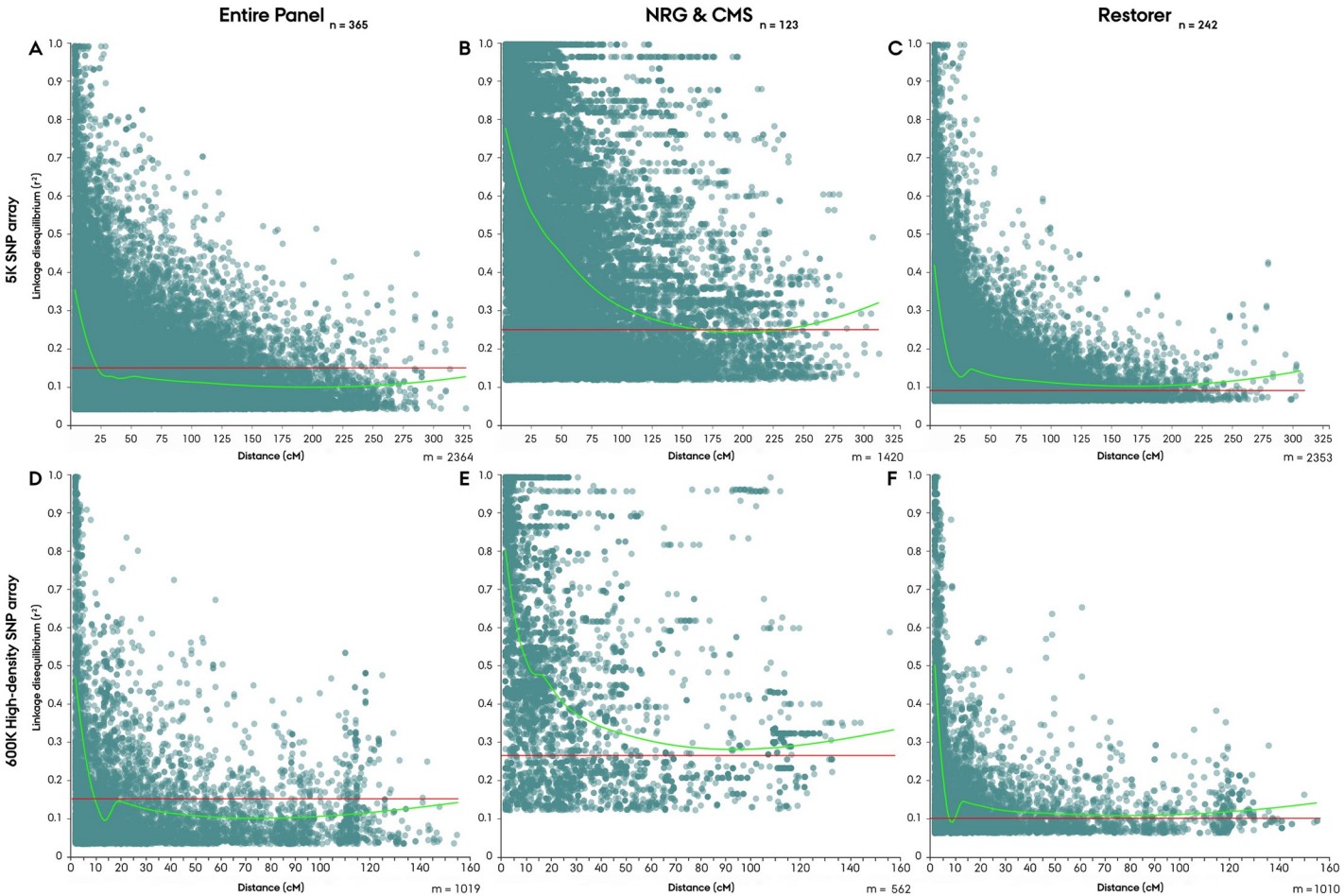

**Fig 8.** Genome-wise linkage decay in the Nordic Seed hybrid rye elite breeding germplasm (**A,D**), seed mother (NRG&CMS) (**B,E**), and, pollen father (Restorer) (**C,F**) population using population specific polymorphic SNP markers from a 5K (**A-C**) and 600K (**D-F**) array. LOESS smooth fitted curve (green) and baseline of population-specific critical value of $r^2$ (red).

The rate of LD decay, determined for the individual arrays, were found to be moderately comparable, with the point of linkage decay determined at 23 cM and 9.5 cM for the 5K-set and 600K-set markers, respectively (Fig 8A and 8D). The rate of decay, however, diverged considerably between the parental populations. In the restorer population, linkage decay could only be determined in the 600K array at 5.5 cM (Fig 8C and 8F), while in the NRG&CMS population it could only be determined in the 5K array at 160 cM (Fig 8B and 8E).

## Discussion

Recent years technological and scientific advances have progressively expedited the introduction of genomic-based techniques in modern plant breeding facilitating an unprecedented insight into the genetic features shaping breeding germplasms [46]. Comprehensive understanding of these features is a necessity to efficiently exploit and continuously advance the genetic potential residing in breeding germplasm to ensure genetic gains and trait-diversity to meet future demands [47]. Currently, DArT, SSR and SNP molecular marker systems have been validated for population studies in the *Secale* genus, OPVs, landraces, hybrid cultivars and small scale inbred populations [1, 25, 48]. In this investigation, we present the first

comprehensive population study of an elite germplasm using the Gülzow-type fertilization control system for hybrid breeding comprising 376 inbred component lines based on 4419 SNP markers.

## Verifying genetic separation of parental populations

Existence of heterotic parental gene pools constitutes the cornerstone in hybrid breeding programs as the prerequisite for achieving a high heterosis effect in hybrid crosses. Our primary objective was, therefore, to confirm and quantify the genetic separation of parental populations. The immediate necessity of a marker-assisted assessment was further emphasized due to partial pedigree information on the NRG&CMS population.

Initial visualization of the germplasm's genetic architecture through PCA and phylogenetic analysis provided an early indication of a distinct genetic separation (Figs 3 and 4). The admixture model of inferred ancestry likewise separated the two parental populations suggesting the existence of two genetically distinct ancestral founder gene pools with limited degree of co-ancestry (Fig 5B and 5G). This infers that the NRG&CMS population is derived from a gene pool genetically distinct to that of the 'Petkus' and 'Carsten'. Estimation of the fixation indices ($F_{ST}$ = 0.332) quantification the genetic separation providing a measure for comparative interpretation. In an extensive population study by Van Inghelandt *et al.* [49] on 1537 inbred hybrid maize elite breeding lines belonging to four heterotic pools, the $F_{ST}$ ranged from 0.06 to 0.29. Bauer *et al.* [1] conducted a similar small-scale population study in an elite hybrid rye breeding germplasm to validate the 600K high-density SNP array. In their study, they confirmed a 'strong' differentiation between parental populations exhibiting an $F_{ST}$ value of 0.229. In conjunction, our study suggests a comparatively high genetic differentiation between the parental populations in the assayed germplasm validating a heterotic pattern suitable for hybrid development. Furthermore, in maize, the relationship between panmictic midparent heterosis, specific combining ability and yielding characteristics has been reported to be positively correlated with the interparental genetic distance [50, 51]. Genetic distance can hence be readily implemented as a simple predictor of hybrid performance in parental crosses prior to development and implementation of more accurate genomic selection procedures [18, 52]. We have confirmed similar positive correlations between interparental genetic distance and average yield performance of hybrids in the germplasm (P. Sarup, personal communication). However, Frisch *et al.* [53] has demonstrated that transcriptome-based distance measure yields more accurate prediction models of hybrid performance.

## Genetic diversity & architectural complexity

Hybrid breeding in rye relies on a three-way cross system involving a genetically distinct pollen parent and a seed parent NRG&CMS. Introgression of foreign released OPVs is a common source for introducing novel genetic variance in otherwise secluded rye breeding germplasm. In practical terms, this involves three-stages, (1) recurrent self-fertilization of OPVs to stabilize their genetic profile, (2) field- and/or greenhouse trials for phenotyping of agronomically important traits, and, (3) evaluation of the genetic compatibility of selected OPVs for introgression in either of the parental gene pools. Restoration of the G-type male sterility (MS) system employed in the CMS population is commonly observed by foreign OPVs, whereas the capacity to maintain the G-type MS, *i.e.* NRG lines, are less frequent [54]. Majority of foreign OPVs introduced to the germplasm has therefore consequently been restorers, hence contributing to an uneven broadening of the germplasms genetic base disfavoring the NRG&CMS population.

As a practical derivative of the three-way cross hybrid system, efforts have primarily been centered around the restorer population relative to the two-component NRG&CMS

population. It is reasonable to presume that these circumstances in conjunction have left a considerable mark on the comparative difference in genetic diversity and architectural complexity between the parental populations. With a sound understanding of the practical breeding history, our study set out to investigate and quantify the derived effect on the genetic features of the elite hybrid rye breeding germplasm. Initial assessment of the breeding lines genetic characteristics led to the finding of a similar low level of residual heterozygosity, high inbreeding coefficient and proportionate effective population size ($N_e/N$) in the two parental populations suggesting exceedingly fixed inbred individuals. While portraying a similar proportionate effective population size it is important to notice the considerable size difference in the actual effective population ($N_e$) with only 76 individuals relative to 137 in the NRG&CMS and restorer population, respectively. Owing to a small effective population size the NRG&CMS population is hence exposed to a strong potential effect of genetic drift inferred to facilitate a loss of genetic diversity over time by random fixation of neutral alleles [55].

Estimation of the within population gene diversity term ($H_s$) confirmed a relative higher level of genetic diversity in the restorer (0.333) relative to the NRG&CMS population (0.250) as expected from the practical breeding history. This divergence in relative genetic diversity could readily be observed in the PCA and phylogenetic analysis providing visualization for a more intuitive interpretation of the estimated genetic characteristics (Figs 3 and 4). In the population study by Bauer *et al.* [1], they identified a similar level of genetic diversity (0.311) within a smaller inbred restorer population, while contrary to our observations a comparatively higher level (0.327) within the NRG&CMS population. In hybrid rye breeding germplasms relying on the more commonly employed 'Pampa' (P) type MS system, introgressed OPVs are contrary to the G type less frequently capable of restoring fertility [14, 56]. As a result, OPVs often exerting the capacity to maintain the P type MS hence contribute to the broadening of the NRG&CMS populations genetic profile. This emphasizes the direct implications of the employed MS system on the introgression of foreign material for broadening the genetic base and the derived effects on the genetic diversity and architectural structure on parental gene pools in elite hybrid rye breeding germplasms. In comparison to elite hybrid breeding germplasm of related cereal species, a similar level of genetic diversity have been reported in maize populations (0.256–0.355) [57] and rice accessions (0.290) [58]. In conjunction, these results indicated a population-wise discrepancy in the genetic diversity defined by a moderate-to-high level of genetic diversity in the restorer population relative to the more genetically narrow NRG&CMS population.

Effects of the practical breeding history could likewise be observed in the parental population's structural architecture. Comparatively, fewer lines could be assigned to secluded subpopulations of a single ancestral component in the restorer population suggesting a more structurally complex admixture which could be explained by the continuous introgression of foreign restorers (Table 1). Interestingly, the Evanno's DeltaK method indicated an additional restorer founder gene pool distinct to the 'Carsten' and 'Petkus' (Fig 5I). Based on available pedigree information, we propose that the more secluded SP2 component is likely derived from the novel ancestral gene pool introduced through introgressed restorers (Table 2, Fig 5I, Teal). The Evanno's DeltaK method, furthermore, suggested that the NRG&CMS population is derived from three, however, unknown ancestral founder populations, portraying a more distinct subpopulation structure (Fig 5D).

## Linkage disequilibrium & decay

Existence of non-random association of alleles at different loci, *i.e.* linkage disequilibrium (LD) is a prerequisite of association genetics facilitating the statistical establishment of

causality between marker and a trait of interest [59, 60]. To determine the population-wise LD, the identified 3383 mapped markers were filtered for population-specific monomorphism causing a considerable discrepancy in marker density between the restorer (3298) and NRG&CMS (1983) population (Table 2). The estimation of genome-wide LD led to the discovery of a large divergence in LD between the parental populations exhibiting a mean of 0.381 and 0.166 in the NRG&CMS and restorer population, respectively. Visualization of the intra-chromosomal pairwise LD by heatmap revealed that the excess of LD in the NRG&CMS population was centered around larger LD blocks spanning beyond the low-recombining peri-centromeric region (Fig 7A, 7C and 7E) [61]. This region is shaped by an abundance of repetitive transposable elements and low-recombination frequency, rendering genes largely inaccessible to breeders [61, 62]. In barley, the peri-centromeric region constitutes 48% of the genome harboring an estimated 14–22% of the gene content [63]. Effect of low-recombination frequency could readily be observed in the 5K-set markers with *e.g.* mean pairwise LD in the NRG&CMS population reaching as high as 0.96 in the 105–110 cM bin on the 7R chromosome (Fig 7E). LD blocks, however, residing outside of the apparent centromeric region provides evidence of chromosomal segments targeted by selection in the germplasm [60]. Conserved blocks in both parental populations suggest a common selection scheme of either an agronomic important trait *e.g.* disease resistance or a domestication-related trait *e.g.* grain shattering [64, 65]. Comparative analysis of the population-wise LD patterns led to the discovery of a common excess on the 3R chromosome in both populations (Fig 6). On the short-arm of 3R chromosome we identified a vast LD block spanning in the NRG&CMS population from 76 to 102 cM on the 5K array (S1 Fig). Several agronomic important QTLs controlling genes affecting preharvest sprouting and drought tolerance have been reported on the 3R chromosome in rye [66, 67]. Melz and Adolf [28], furthermore identified a male sterility (ms) minor gene *ms2* on the 3RL chromosome, later denoted as 'restorer of fertility' (*Rf*) 'Gülzow' type *Rfg2* which could likewise be related to the conservation of this segment in the germplasm [28, 68]. While functional analysis in rye is impeded by poor or insufficient gene annotation, certain agronomic traits *e.g.* male-fertility restoration is, however, well annotated, with the major *Rf* genes of the 'Pampa' type, *Rfp1*, *Rfp2*, and, *Rfp3* accurately mapped to the 4RL chromosome [69, 70]. Intriguingly we identified a strongly conserved LD block in both parental populations at the distal region of the 4RL chromosome, with 9/12 markers in the restorer population and 3/5 markers in the NRG&CMS population annotated as '*Rfp3*'. The 'Gülzow' *Rfg1* and 'C' type *Rfc1 Rf* major genes have likewise been mapped to the distal region of the 4RL chromosome and been proposed allelic to the P-type [68, 71]. The novel rye reference genome by Rabanus-Wallace *et al.* [21], however, constitutes a milestone in large-scale functional analysis, *de novo* annotating the near-full complement of 34.441 gene models in the rye genome.

As an outcrossing crop species rye has been reported to exhibit a rapid rate of LD decay [72]. In our analysis the point of LD decay set as the interception point of the LOESS regression curve and the population-specific critical value of $r^2$ could, however, not be determined for the NRG&CMS population (Fig 8B and 8E). This finding is likely related to the substantial intrachromosomal LD structure observed within the population, impeding the standard decay of linkage as a function of the inter-loci distance [73]. The high LD within the NRG&CMS population putatively derives from a low frequency of detectable recombination events caused by a depletion of rare alleles within the NRG&CMS population with ~50% of the polymorphic markers exhibiting a MAF $\leq 0.1$ (Fig 1A) [74]. In aggregate, these findings suggest the occurrence of a demographic bottleneck event or intense selection in the NRG&CMS population supported by the observations of a lower genetic diversity, structural complexity and effective population size [75]. It is possible that the low frequency of lines exhibiting the ability to maintain the employed G-type MS system (*i.e.* NRG lines) could have exerted a considerable

selection pressure with introgressed lines exhibiting a high degree of relatedness. In the restorer population, which portrayed a more homogeneous LD structure with minor LD blocks, linkage decay could be determined in the 600K SNP array at 5.5 cM (Figs 7B, 7D and 8F). By comparison, in a study on an elite hybrid maize breeding germplasm by Van Inghelandt *et al.* [76], LD decay was observed in the range from 0.11 to 2.74 cM across four heterotic gene pools. The comparatively rapid rate of LD decay in outcrossing species such as rye promises high resolution for association studies while requiring an equivalently higher marker density [72].

### Introgression of novel genetic variation from foreign sources

Genetic diversity constitutes the fabric of which novel cultivars are shaped, portraying a unique combination of alleles superior to the desired purpose. Erosion of genetic diversity, therefore, constitutes a serious threat to modern plant breeding with loss of features imperative to meet the future demands [77]. Findings of our study emphasizes foremost the necessity of addressing the strong population structure and narrow genetic profile of the NRG&CMS population to sustain the genetic gains in the hybrid rye breeding program. The comparatively small effective population size of both parental populations furthermore emphasizes the need for broadening the germplasm, particularly in the NRG&CMS population. In rye, several population genetics studies have recently investigated the accessible genetic reservoir residing in landraces, OPVs, and wild *Secale* species for this immediate purpose [3, 26, 77]. Introgression of landraces and OPVs in elite hybrid breeding germplasms require a comprehensive evaluation of their genetic compatibility in order to preserve the genetic separation of parental gene pools [78]. Marker assisted introgression of OPVs has successfully been demonstrated by Fischer *et al.* [16] in a ´Petkus´ and ´Carsten´ based elite hybrid rye breeding germplasm. Introgression libraries of an exotic Iranian landrace has likewise been developed by Falke *et al.* [79], and demonstrated by marker-assisted backcrossing to significantly enhance baking quality traits relative to the recurrent parent [80].

### Performance and informativeness of SNP markers

Stringent assessment of marker properties and informativeness constitutes an imperative step in the successful application of genomic-based breeding techniques in plant breeding. Analysis of polymorphic information content (PIC) is often used as a measure of the informativeness of a genetic marker for linkage studies but has likewise been deployed as a genetic diversity term [81]. Across marker-sets, the 5K-set and 600K-set were highly similar in terms of mean marker PIC (0.268, 0.262) and mean MAF (0.254, 0.250) score. The smaller wheat-set markers (426), however, exhibited a slightly lower PIC (0.225) and MAF (0.201) score. In our study, we determined a mean PIC value of 0.20 and 0.26 across the population-specific polymorphic markers in the NRG&CMS and restorer population, respectively (Fig 1B). These values correspond to PIC values reported in SNP-based population studies on elite hybrid breeding germplasms in maize (0.29) [57] and rice (0.23) [58]. Amongst the 4419 markers, we discovered a considerable divergence in population-specific polymorphic markers with 1870 and 38 monomorphic markers in the NRG&CMS and restorer population, respectively. Initially this divergence was interpreted as a bias introduced in the quality filtration step as a product of the population size difference. Population-wise quality filtration on the entire marker panel, however, only led to the identification of an additional 271 low-polymorphism markers for the NRG&CMS population. This can either be explained by the narrow genetic profile of the NRG&CMS population characterized by an depletion of rare alleles due to higher genetic drift (Fig 1A) and/or by a diverging pattern in SNPs relative to the parental lines utilized for the 5K and 600K SNP array

design more closely resembling the restorer population [1, 20]. If the latter explanation is correct this further supports the distinct genetic separation of the parental populations, characterized by a unique population-dependent pattern in SNPs.

In our investigation of LD structure and rate of decay, the separation of markers into a 5K-set and 600K-set due to diverging array distance measure likewise enabled a comparative assessment of the marker sets performance and informativeness. It is important to emphasize that with an incomplete subset of markers from the 5K an 600K SNP array we cannot speculate on their comparative performance and informativeness. Neither of the assessed genetic characteristics including LD, intrachromosomal organization of LD, and, progression of linkage decay was found to consistently diverge between the marker sets in our analysis (Table 2, Figs 7 and 8). Genetic mapping of the SNPs likewise revealed a similar positional pattern in the arrays, displaying an evident excess of markers in the peri-centromeric region (Fig 2A and 2B). While the non-uniform intrachromosomal distribution of markers clustering in the peri-centromeric region, pose question to the informativeness of the SNP marker panel we observed no apparent evidence of a positional trend on the assessed genetic characteristics. Conclusively, our analysis found no evidence of divergence in performance nor informativeness between the 5K-set and 600K-set markers, hence endorsing the 20K SNP chip construct.

## Conclusion

In the present study, we demonstrated the application of a SNP-based marker system for dissecting the genetics of a large elite hybrid rye breeding germplasm. Through a palette of complimenting analysis, we confirmed a strong genetic differentiation of parental populations. These populations were found to diverge in several features with the NRG&CMS population portraying a strong population structure characterized by a narrow genetic profile, small effective population size and high genome-wise LD. We propose that employed MS system putatively constitutes a population determining parameter by influencing the rate of introducing novel genetic variation. Functional analysis of LD blocks for inference of selection on agronomic important traits in the germplasm led to the finding of a conserved segment on the distal 4RL chromosomal region annotated to the *Rfp3* male-fertility restoration gene.

Considering the plethora of diversity preserved in genetic resources it is the emphasis of this study to pursue a marker-assisted broadening of the NRG&CMS population to address the strong population structure and narrow genetic profile. Furthermore, the novel rye reference genome by Rabanus-Wallace *et al.* [21] facilitates physical mapping and annotation of the 5K-set and 600K-set markers for combined linkage study and large-scale functional analysis to further unravel the genetic features of the hybrid rye elite breeding germplasm.

## Supporting information

**S1 Table. Opposing population ancestry from an admixture model of inferred ancestry at K set to 2 and observed residual heterozygosity (Ho) of 11 discarded Nordic Seed hybrid rye elite breeding lines belonging to the restorer (R) and non-restorer germplasm (NRG) population.**
(DOCX)

**S1 Fig. Pairwise intra-chromosomal linkage disequilibrium (LD, r2) in the Nordic Seed hybrid rye elite breeding populations using population-specific polymorphic SNP markers originating from a 5K (mNRG&CMS = 1420, mR = 2353) and 600K (mNRG&CMS = 562, mR = 1010) array visualized by heatmap for the seed mother (NRG&CMS) and pollen**

**father (R) population.**
(TIF)

**S1 File.**
(ZIP)

## Acknowledgments

We would like to thank laboratory technician Hanne Svenstrup at Nordic Seed A/S for contribution to the genotypic data collection. The plant material included in this study was provided by Nordic Seed Germany GmbH (Nienstädt, Germany). The high throughput SNP genotyping was performed by Trait Genetics (Gatersleben, Germany).

## Author Contributions

**Conceptualization:** Nikolaj M. Vendelbo, Jihad Orabi, Ahmed Jahoor.

**Data curation:** Nikolaj M. Vendelbo, Pernille Sarup.

**Formal analysis:** Nikolaj M. Vendelbo, Pernille Sarup, Jihad Orabi.

**Funding acquisition:** Jihad Orabi, Ahmed Jahoor.

**Investigation:** Nikolaj M. Vendelbo, Peter S. Kristensen, Ahmed Jahoor.

**Methodology:** Nikolaj M. Vendelbo, Ahmed Jahoor.

**Project administration:** Jihad Orabi, Ahmed Jahoor.

**Resources:** Jihad Orabi, Peter S. Kristensen.

**Software:** Nikolaj M. Vendelbo, Pernille Sarup.

**Supervision:** Pernille Sarup, Jihad Orabi, Ahmed Jahoor.

**Validation:** Pernille Sarup, Jihad Orabi, Peter S. Kristensen, Ahmed Jahoor.

**Visualization:** Nikolaj M. Vendelbo.

**Writing – original draft:** Nikolaj M. Vendelbo.

**Writing – review & editing:** Nikolaj M. Vendelbo, Pernille Sarup, Jihad Orabi, Peter S. Kristensen, Ahmed Jahoor.

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
