## [Decision Letter · Decision Letter 0]

8 Jul 2020

PONE-D-20-14942

Dissecting the Genetics of a Hybrid Rye Breeding Germplasm

PLOS ONE

Dear Dr. Vendelbo,

Thank you for submitting your manuscript to PLOS ONE. After careful consideration, we feel that it has merit but does not fully meet PLOS ONE’s publication criteria as it currently stands. Therefore, we invite you to submit a revised version of the manuscript that addresses the points raised during the review process.

I received comments from the advisers on your manuscript "Dissecting the Genetics of a Hybrid Rye Breeding Germplasm" which you submitted to PlosONE. Based on reviewer comments and my assessment I have decided that your manuscript could be reconsidered for publication should you be prepared to incorporate major revisions. When preparing revised manuscript, you are asked to carefully consider the reviewer comments which can be found below, and submit a list of detailed and itemized responses to the comments.

We look forward to receiving your revised manuscript.

Kind regards,

Dragan Perovic, Ph.D

Academic Editor

PLOS ONE

Additional Editor Comments:

Dear Dr. Meisner Vendelbo,

I received comments from the advisers on your manuscript "Dissecting the Genetics of a Hybrid Rye Breeding Germplasm" which you submitted to PlosONE. Based on reviewer comments and my assessment I have decided that your manuscript could be reconsidered for publication should you be prepared to incorporate major revisions. When preparing revised manuscript, you are asked to carefully consider the reviewer comments which can be found below, and submit a list of detailed and itemized responses to the comments.

With kind regards,

Dragan Perovic

Journal Requirements:

"The research was funded by Innovation Fund Denmark (grant no. 8053-00085B), Pajbjergfoundation, and Nordic Seed A/S. The funders had no role in study design, data collection and analysis, decision to publish, or preparation of the manuscript. "

We note that one or more of the authors have an affiliation to the commercial funders of this research study : Nordic Seed A/S.

2.1. Please provide an amended Funding Statement declaring this commercial affiliation, as well as a statement regarding the Role of Funders in your study. If the funding organization did not play a role in the study design, data collection and analysis, decision to publish, or preparation of the manuscript and only provided financial support in the form of authors' salaries and/or research materials, please review your statements relating to the author contributions, and ensure you have specifically and accurately indicated the role(s) that these authors had in your study. You can update author roles in the Author Contributions section of the online submission form.

2.2. Please also provide an updated Competing Interests Statement declaring this commercial affiliation along with any other relevant declarations relating to employment, consultancy, patents, products in development, or marketed products, etc. 

Reviewers' comments:

Reviewer's Responses to Questions

**Comments to the Author**

1. Is the manuscript technically sound, and do the data support the conclusions?

Reviewer #1: Yes

Reviewer #2: Partly

2. Has the statistical analysis been performed appropriately and rigorously? 

Reviewer #1: Yes

Reviewer #2: Yes

3. Have the authors made all data underlying the findings in their manuscript fully available?

Reviewer #1: Yes

Reviewer #2: No

4. Is the manuscript presented in an intelligible fashion and written in standard English?

Reviewer #1: Yes

Reviewer #2: Yes

5. Review Comments to the Author

Reviewer #1: Comments to the authors are attached in a WORD file; further corrections are made on the pdf-file:

# Title should be modified: “Genetic structure of a germplasm for hybrid breeding in rye”

# Figure 7a & b could be removed.

# Please, format „References“ in a uniform manner, e.g. author names

# Italics for species names

# Complete list of authors under “References”

Reviewer #2: The manuscript reports on a comprehensive population genetic analysis of elite germplasm used for hybrid breeding in rye (Secale cereale L.). Hybrid breeding is a cutting edge breeding method with as a crucial role in improving global food security and helping to meet the ambitious production targets for 2050 (Whitford et al. 2013). Hybrid breeding keeps rye competitive in modern agricultural production systems since more than four decades, with the Pampa- (P-) type CMS being the pre-dominating hybridization system. In contrast, the Gülzow- (G-) type CMS is used for hybrid breeding in the elite germplasm in the current study. Thus, the breeding program is an invaluable contribution to reduce the genetic vulnerability (cf. Lewings 1990) of hybrid rye and the manuscript a highly information piece of research and very useful share to the knowledge about the subject, which is within the scope of the journal.

The objectives of the study are clear and almost appropriate. As precisely outlined by the authors (l.77), Bauer et al. (2017) already reported on a SNP-based population study of a hybrid rye breeding population. Although the number of genotypes used by Bauer et al. (loc. cit.) was lower as compared to the present study and included individuals from open‐pollinated genetic resources, the number of SNP markers used to characterize the material was more than 50x higher as compared to the present study. It is, thus, misleading and not appropriate to claim the first comprehensive SNP-based population study of a large hybrid breeding population (l.16, l.79, l.348). This needs to be revised. In fact, you report the first comprehensive SNP-based population study of elite germplasm using a fertilization control system for hybrid breeding, that is genetically different to the pre-dominating P-type.

The question posed by the authors is well defined and the applied molecular and statistic methods are appropriate and well described. Overall, the data are sound and the manuscript adheres to the relevant standards for reporting and data deposition except of l.164-165: ‘The test yielded an additional 271 high-quality markers for the NRG&CMS population, while only 4 for the restorer population.’ These markers need to be specified in detail as supporting information - everything necessary to reproduce the work needs to be available. For a paper that is largely computational, a CSV repository and at least minimal README is the standard.

The discussion and conclusions are quite well balanced and adequately supported by the data except of l.371-372: ‘Genetic distance can hence be readily implemented as simple predictor of hybrid performance…’. Prediction methods using the genetic distance between the parental lines failed to reliably predict the hybrid performance of inter-pool hybrids in plant breeding programs (Melchinger 1999). Rather, prediction of hybrid performance with transcriptome-based distances using selected markers proofed to be more precise than prediction models using DNA markers or general combining ability estimates using field data (Frisch et al. 2010).

The manuscript is well-written, but some given statements appear to be not logical: in l.176 the sentence is incomplete, none of the wheat-set markers were mapped in rye. L.262 & l377-378: use the term parent instead of ‘mother’ and ‘father’. There is little or no connection between the sentence in l.463-465 to the paragraph given before.

The literature cited is generally thoughtful and focused, but not completly up-to-date (l.498). Falke et al. (2008, 2009) investigated the suitability of a landrace/primitive rye for broadening the genetic base of improved hybrid rye germplasm

To conclude, the manuscript is recommended for publication in PONE after major revisions have been performed.

References

Bauer E, Schmutzer T, Barilar I, Mascher M, Gundlach H, Martis MM, Twardziok SO, Hackauf B, Gordillo A, Wilde P, Schmidt M, Korzun V, Mayer KF, Schmid K, Schön CC, Scholz U. (2017) Towards a whole-genome sequence for rye (Secale cereale L.). Plant J. 89:853-869. doi: 10.1111/tpj.13436.

Falke KC, Susić Z, Hackauf B, Korzun V, Schondelmaier J, Wilde P, Wehling P, Wortmann H, Mank R, Rouppe van der Voort J, Maurer HP, Miedaner T, Geiger HH (2008) Establishment of introgression libraries in hybrid rye (Secale cereale L.) from an Iranian primitive accession as a new tool for rye breeding and genomics. Theor Appl Genet 117:641-652. doi: 10.1007/s00122-008-0808-1

Falke KC, Susić Z, Wilde P, Wortmann H, Möhring J, Piepho HP, Geiger HH, Miedaner T (2009) Testcross performance of rye introgression lines developed by marker-assisted backcrossing using an Iranian accession as donor. Theor Appl Genet 118:1225-1238. doi: 10.1007/s00122-009-0976-7

Matthias Frisch, Alexander Thiemann, Junjie Fu, Tobias A Schrag, Stefan Scholten, Albrecht E Melchinger (2010) Transcriptome-based Distance Measures for Grouping of Germplasm and Prediction of Hybrid Performance in Maize Theor Appl Genet 120:441-450. doi: 10.1007/s00122-009-1204-1.

Levings CS 3rd (1990) The Texas Cytoplasm of Maize: Cytoplasmic Male Sterility and Disease Susceptibility. Science 250:942-947. doi: 10.1126/science.250.4983.942.

Melchinger AE (1999) Genetic diversity and heterosis. In: Coors JG, Pandey S (eds) The genetics and exploitation of heterosis in crops. ASA-CSSA, Madison, pp 99–118

6. PLOS authors have the option to publish the peer review history of their article (what does this mean?). If published, this will include your full peer review and any attached files.

Reviewer #1: No

Reviewer #2: No

---

## [Author Response · Author response to Decision Letter 0]

28 Jul 2020

Dear reviewers and editors, 

Thank you for your comments, we have carefully and thoroughly read through these and addressed them accordingly. You can find a marking for each of the corrections in the marked manuscript file, with a specified 'code' to where a detailed account of the correction can be found in the 'response to reviewers' letter. We hope that we have addressed all of the issues that you have so keenly identified. 

Sincerely

The authors

---

## [Decision Letter · Decision Letter 1]

27 Aug 2020

PONE-D-20-14942R1

Genetic Structure of a Germplasm for Hybrid Breeding in Rye

PLOS ONE

Dear Dr. Vendelbo,

Thank you for submitting your manuscript to PLOS ONE. After careful consideration, we feel that it has merit but does not fully meet PLOS ONE’s publication criteria as it currently stands. Therefore, we invite you to submit a revised version of the manuscript that addresses the points raised during the review process.

Dear Dr. Meisner Vendelbo,

I received comments from the advisers on revised version of your manuscript "Dissecting the Genetics of a Hybrid Rye Breeding Germplasm" submitted to PlosONE. Major obstacle in the current version of the manuscript is the data availability. Please see comments of reviewer 2. Therefore, your manuscript could be reconsidered for publication should you be prepared to incorporate minor revisions by providing the data in a requested format.

With kind regards,

Dragan Perovic

We look forward to receiving your revised manuscript.

Kind regards,

Dragan Perovic, Ph.D

Academic Editor

PLOS ONE

Additional Editor Comments (if provided):

Dear Dr. Meisner Vendelbo,

I received comments from the advisers on revised version of your manuscript "Dissecting the Genetics of a Hybrid Rye Breeding Germplasm" submitted to PlosONE. Major obstacle in the current version of the manuscript is the data availability. Please see comments of reviewer 2. Therefore, your manuscript could be reconsidered for publication should you be prepared to incorporate minor revisions by providing the data in a requested format.

With kind regards,

Dragan Perovic

Reviewers' comments:

Reviewer's Responses to Questions

**Comments to the Author**

1. If the authors have adequately addressed your comments raised in a previous round of review and you feel that this manuscript is now acceptable for publication, you may indicate that here to bypass the “Comments to the Author” section, enter your conflict of interest statement in the “Confidential to Editor” section, and submit your "Accept" recommendation.

Reviewer #1: All comments have been addressed

Reviewer #2: All comments have been addressed

2. Is the manuscript technically sound, and do the data support the conclusions?

Reviewer #1: Yes

Reviewer #2: Yes

3. Has the statistical analysis been performed appropriately and rigorously? 

Reviewer #1: Yes

Reviewer #2: Yes

4. Have the authors made all data underlying the findings in their manuscript fully available?

Reviewer #1: Yes

Reviewer #2: No

5. Is the manuscript presented in an intelligible fashion and written in standard English?

Reviewer #1: Yes

Reviewer #2: Yes

6. Review Comments to the Author

Reviewer #1: Revision made meets the reviewers intention. No additional changes are required. You may prepare the finald version.

Reviewer #2: Unfortunatly, you have not adequately addressed my comments raised in the first round of review concerning data availability.

As PLOS believes that sharing data fosters scientific progress, PLOS journals require authors to make all data necessary to replicate their study’s findings publicly available WITHOUT RESTRICTION at the time of publication. The anonymised marker IDs as given in the CSV repository, that you have added in the revised version of your manuscript, do not comply with PLOS policy on data availability. While the provided information indeed enables a reanalysis of your data or exploit the dataset in a new analysis, it is still impossible to validate, replicate, reproduce, include into meta-analyses and/or reinterprete your research.

A publication is conditional on compliance with the PLOS policy on data availability. Thus, it is mandatory that all markers in the CSV repository need to be given with their IDs of the custom SNP array. Furthermore, the 271 high-quality markers for the NRG&CMS population as well as the 4 markers identified for the restorer population needs to be specified in an appropriate manner.

7. PLOS authors have the option to publish the peer review history of their article (what does this mean?). If published, this will include your full peer review and any attached files.

Reviewer #1: No

Reviewer #2: No

---

## [Author Response · Author response to Decision Letter 1]

1 Sep 2020

Dear reviewers, you can find my comments in the attached "Response to reviewer" document

---

## [Decision Letter · Decision Letter 2]

9 Sep 2020

Genetic Structure of a Germplasm for Hybrid Breeding in Rye

PONE-D-20-14942R2

Dear Dr. Vendelbo,

We’re pleased to inform you that your manuscript has been judged scientifically suitable for publication and will be formally accepted for publication once it meets all outstanding technical requirements.

Kind regards,

Dragan Perovic, Ph.D

Academic Editor

PLOS ONE

Additional Editor Comments (optional):

Dear Dr Vendelebo,

it is my pleasure to accept your article to be published at PlosONE.

Regards

Dragan

Reviewers' comments:

Reviewer's Responses to Questions

**Comments to the Author**

1. If the authors have adequately addressed your comments raised in a previous round of review and you feel that this manuscript is now acceptable for publication, you may indicate that here to bypass the “Comments to the Author” section, enter your conflict of interest statement in the “Confidential to Editor” section, and submit your "Accept" recommendation.

Reviewer #2: All comments have been addressed

2. Is the manuscript technically sound, and do the data support the conclusions?

Reviewer #2: Yes

3. Has the statistical analysis been performed appropriately and rigorously? 

Reviewer #2: Yes

4. Have the authors made all data underlying the findings in their manuscript fully available?

Reviewer #2: Yes

5. Is the manuscript presented in an intelligible fashion and written in standard English?

Reviewer #2: Yes

6. Review Comments to the Author

Reviewer #2: All markers in the CSV repository are given with their IDs of the custom SNP array. Furthermore, the 271 high-quality markers for the NRG&CMS population as well as the 4 markers identified for the restorer population are specified in an appropriate manner. The provided information indeed enables a reanalysis of the data or exploit the dataset in a new analysis. In addition its now possible to validate, replicate, reproduce, include into meta-analyses and/or reinterprete the research. To conclude, the revised version complies with the PLOS policy on data availability and can be recommended for publication in Plos One.

7. PLOS authors have the option to publish the peer review history of their article (what does this mean?). If published, this will include your full peer review and any attached files.

Reviewer #2: No

---

## [Editor Report · Acceptance letter]

14 Sep 2020

PONE-D-20-14942R2 

Genetic Structure of a Germplasm for Hybrid Breeding in Rye  

Dear Dr. Vendelbo:

I'm pleased to inform you that your manuscript has been deemed suitable for publication in PLOS ONE. Congratulations! Your manuscript is now with our production department. 

Kind regards, 

on behalf of

Dr. Dragan Perovic 

Academic Editor

PLOS ONE